# Host-pathogen protein interaction studies: quality control of cDNA libraries using nanopore sequencing

Cécile Schimmich[1], Mathilde Gondard[2], Gregory Caignard[2],
José-Carlos Valle-Casuso[1,3], Damien Vitour[2], François Piumi[2]*

1 Unité physiopathologie et épidémiologie des maladies équines (PhEED), Laboratoire de Santé Animale, ANSES, Goustranville, France, 2 UMR Virologie, INRAE, École nationale vétérinaire d'Alfort, Anses, Laboratoire de Santé animale, Université Paris-Est, Maisons-Alfort, France, 3 Mixed Technological Unit "Equine Health and Welfare – Organisation and Traceability of the Equine Industry" (UMT SABOT), Goustranville, France

* francois.piumi@vet-alfort.fr

## Abstract

Protein-protein interactions (PPI) play a key role in host-pathogens interaction studies, as proteins are essential to many cellular mechanisms. The yeast two-hybrid (Y2H) approach is a well-established method for high-throughput PPI screening and mapping of protein interaction networks. The success of this approach partially depends on the quality and representativeness of the host cDNA library, which can be constructed from the transcriptomic content of a selected host cellular type. However, evaluating the relevance of the cDNA library content remains challenging, and one of the key limitations of this interactomic approach is the occurrence of false-negative results (i.e., the absence of detectable interactions). Here, we report a direct, long read, high-throughput sequencing method using Oxford Nanopore Technologies, to assess the completeness of the host cDNA library used in host-pathogen interactions Y2H screening. This approach enables easy identification of possible downstream screened genes in PPI assays, minimizing sequencing biases and bioinformatics handling of the data. This study was performed on a cDNA library, generated from A549 human lung carcinoma cells. We were able to identify 12,123 protein coding genes from the sequencing of whole plasmids containing the cDNA inserts, that were further analyzed via functional pathways enrichment for deeper characterization. This diversity and relative abundance evaluation method could be a first step when generating new cDNA libraries of interest for PPI studies, ensuring the validity and suitability of the host library before proceeding with all Y2H screening steps.

**Data availability statement:** All sequence files are available from the SRA database (accession number: SRR31585952).

**Funding:** The author(s) received no specific funding for this work.

**Competing interests:** The authors have declared that no competing interests exist.

## Introduction

Protein-protein interactions (PPIs) are crucial for cell life cycle, as they are involved in various biological processes and mechanisms. In the regard of viral infections of target cells, interplay between cellular proteins and viral proteins is necessary. The yeast two-hybrid method (Y2H), described over three decades ago [1] is a high-throughput PPI screening method, particularly suitable for virus-host PPIs study. It is an unbiased binary PPI test where defined viral proteins can be tested individually against whole cDNA or DNA libraries, representing host proteins. This approach was used to generate protein interaction networks for various viruses, such as hepatitis C virus [2], vaccinia virus [3] human T-cell leukemia virus [4] or coronaviruses [5]. The Y2H method allows for large-scale putative protein partners screening with a cheap and unbiased approach. One of the main described limitations of this method is the rate of false-negative due to steric hindrance of the fusion construction and a lack of reproducibility [6]. Additionally, the activation of the reporter gene, which is essential for identifying PPIs, occurs in the yeast nucleus. Consequently, proteins with hydrophobic transmembrane domains are unable to enter the nucleus and participate in the screening process [7]. Another limiting factor can directly come from the absence of the possible interactor within the screened cDNA library. One way to improve Y2H high-throughput screening assay using cDNA libraries is to characterize the input partners upstream of the assay itself. Traditional ways of evaluating cDNA library pools, often used for PPI assays such as Y2H, only provide limited information on the library diversity, picking few colonies, tens to hundreds of clones and sequencing the contained inserts [8–11]. Quality indicators of a suitable cDNA library include the cloning efficiency, the length of cDNA inserts, the number of inserted genes or diversity and the gene abundance, meaning the number of copies per represented genes [12]. The importance of a normalized library was also mentioned, to avoid over-representation of over-represented genes and linked biases [9].

This high-throughput sequencing strategy of cDNA libraries was already described in reports [12,13] using different strategies, short-reads sequencing with Illumina or long-reads sequencing with Oxford Nanopore Technologies (ONT). The use of next generation sequencing (NGS) to perform the Y2H assay directly after the yeast mating step, avoiding long selection and yeast subculturing was also reported [14]. This direct sequencing after mating does not account for possible false positives, that can be eliminated through medium selection for a couple of weeks [15]. ONT sequencing of long reads relies on detecting changes in ionic current as negatively charged DNA or RNA molecules pass through a nanopore embedded in a charged membrane. These current changes correspond to different nucleotides, which are then decoded by algorithms performing base calling [16]. Over the decade of its market availability, the ONT sequencing was improved through changes of the nanopore itself, used chemistry, algorithms updates, all improving sequencing length and accuracy [17]. With the always evolving ONT sequencing, the latest chemistry, Kit 14 chemistry combining R10.4.1 flow cells and LSK114 ligation sequencing kit V14 appears to yield very accurate sequences, over 99% accuracy expected, 98.9% reported in a

plasmid sequencing study [18]. The use of ONT long-read sequencing is widely reported to sequence plasmids, in bacterial surveillance [18–20] and as well as a molecular biology tool to verify constructions [21,22].

In this study we report a direct application and efficient strategy of high-throughput sequencing of a Y2H ready cDNA library, generated from A549 human cell line. First, the cDNA inserts containing vectors were linearized using a restriction enzyme. Then ONT sequencing and mapping to the reference genome was performed. We finally propose an easy straightforward bioinformatics pipeline of direct identification of human genes contained within the bacterial backbone. The sequencing saturation curve representing the completeness of the sequencing and functional representability was assessed using Gene Ontology terms analysis as well as Reactome pathways analysis (Fig 1 and Fig S1).

## Materials and Methods

### cDNA library construction

A549 cells (ATCC CCL-185) were grown in DMEM supplemented with 10% heat-inactivated fetal calf serum, 1% non-essential amino acids, and 100 IU/ml penicillin and 100 μg/ml streptomycin. The day before collection, cells were passaged (p89) and split into ten T150 culture flasks. Twenty-four hours later, cells were trypsinized and pelleted by centrifugation before being stored at −80°C until shipment. Finally, cell pellets were sent on dry ice to Life Technologies for

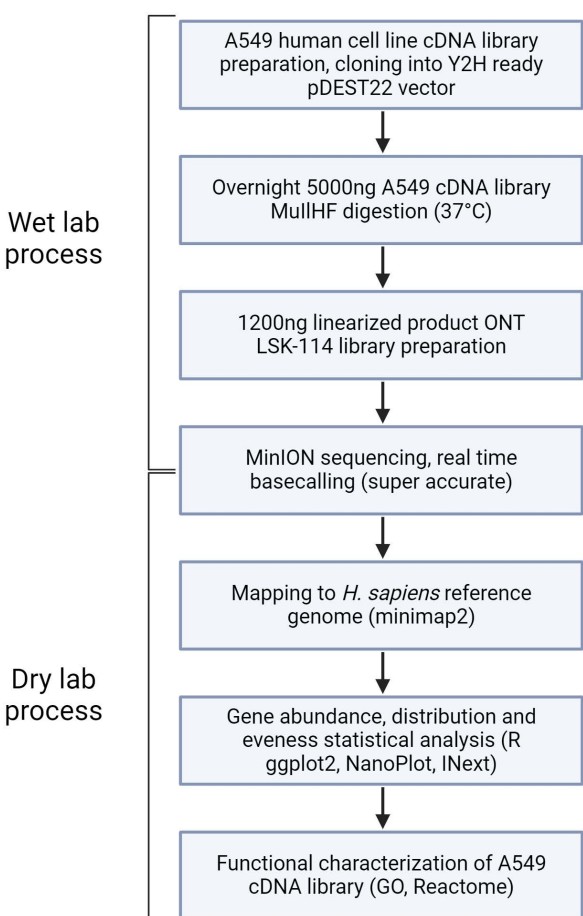

**Fig 1. Overview of the experimental and analytical steps leading to the characterization of the A549 cDNA library by long-read sequencing.**

outsourced cDNA library generation (Carlsbad, CA, USA). Total RNA extraction was performed using Trizol reagent. The library was constructed using Uncut Three Reading Frame procedure (CloneMiner™ technology), which does not employ a restriction enzyme digestion step in the cloning procedure. First strand DNA synthesis was performed using SuperScript III reverse transcriptase and the arrB2-dT22VN primer. Second strand synthesis results in double strand cDNA. attB1 adaptor was ligated to 5' end of cDNA. Then, cDNA with attB flanked ends were cloned into the attP-containing donor vector pENTR222 through site-specific recombination (GATEWAY® technology). The resulting GATEWAY® entry cDNA library is then shuttled (LR recombination reaction) into the GATEWAY® destination vector pDEST22 downstream of the Gal4 activation domain. After electroporation of highly competent DH10 cells, the amplified library is finally obtained after plasmid purification. This cDNA library synthesis was subcontracted, with quality controls checks, including the quality and quantity of RNA material used or the average insert size and the percentage of vectors containing inserts, determined using 24 random clones. The company guarantees an average insert size of over 1kb and that the library will have greater than 87% of vectors with inserts. The generated product, the A549 cDNA library was kindly provided by Dr. Pierre-Olivier Vidalain (CIRI, Lyon, France).

## Oxford Nanopore Technologies library preparation and sequencing

5000 ng of cDNA library were digested overnight at 37°C using the *Mlu*I-HF restriction enzyme (100 U) in order to linearize pDEST22 vectors. There is a unique site of *Mlu*I cutting on the pDEST22, right at the end of the Gal4 activation domain, before the GATEWAY® attR site used to clone the inserts. This restriction site is also unlikely to be found in eukaryotic cDNA inserts sequences. Then 1200 ng of linearized A549 cDNA library was used with the SQK-LSK114 kit (Oxford Nanopore Technologies) and the Ligation sequencing amplicons V14 protocol available on the manufacturer website (Document version: ACDE_9163_v114_revU_29Jun2022), according to the manufacturer's instructions. Sequencing was performed on R10.4.1 flow-cells (FLO-MIN114, Oxford Nanopore Technologies) on a MinION Mk1b device with MinKNOW v.23.11.5 software (Oxford Nanopore Technologies). We used two identically prepared LSK114 libraries loaded onto two R10.4.1 flow-cells to generate the total sequenced reads dataset analyzed in this article.

## Basecalling, quality control of raw reads

Raw ONT data (POD5) were directly basecalled while sequencing with MinKNOW software (v.23.11.5) using the Super-accurate basecalling model (Dorado v.7.2.13) with a minimum quality score set to 10 (Q10) and enabling the simultaneous removing of the sequencing adaptors. Basecalled reads were saved as fastq files (accession number: SRR31585952) and the data from the two sequencing runs were merged to obtain a unique dataset.

## Quality control

Reads quality was assessed using NanoPlot (v1.42.0), specially built for Nanopore sequences. NanoPlot produces summary statistics and plots on data quality.

## Read alignment to the reference genome and transcriptome

Alignments were performed using minimap2 (v2-2.26) [23] on the genome (Ensembl GRCh38 release 110), the transcriptome (Ensembl cDNA GRCh38 release 112) and a concatenation of the transcriptome and the non-coding transcriptome (Ensembl ncRNA GRCh38 release 112) with the following arguments: -ax map-ont –N 100. The –a argument specifies that the alignments should be output in SAM format (Sequence Alignment/Map), a standard format for representing sequence alignments. The -x map-ont preset adjusts minimap2's internal parameters to handle the characteristics of ONT reads, which are long reads (tens to hundreds of thousands of bases long) and which may have a higher error rate compared to other sequencing technologies, such as Illumina. This preset ensures robust alignments despite these errors.

The -N parameter specifies the maximum number of secondary alignments reported per read. In genome alignment, the purpose is to map reads to the entire genomic sequence, which often contains repetitive elements, duplications, and homologous regions. Using -N 100 ensures to capture all possible mappings in a complex reference, especially in repetitive regions. A splice-aware alignment against the reference genome was realized with the following arguments: -ax splice –N 10. For splice-aware alignments with cDNA reads, the goal is typically to align reads to their true genomic origin while considering exon-exon junctions. Using a -N 10 ensures to reduce secondary alignments to focus on relevant mappings and limit alignments to biologically meaningful regions.

### Gene abundance estimation and functional analysis

Alignment files from minimap2 were converted to bam format, sorted and indexed using samtools v1.19 [24]. The occurrences of each transcript were counted in the filtered bam file resulting from the alignments to the reference genome and transcriptome. The iNEXT R package (v3.0.1) [25] was used to calculate the saturation curve.

Functional enrichment analysis was achieved using the clusterProfiler R package (v4.6.0) [26]. Semantic similarity among GO terms was computed with the GOSEmSim R package (v2.24.0) [27]. Enrichment analysis was also implemented using the ReactomePA R package (v1.50.0) [28].

### Code availability

The bioinformatics pipeline described in this article is available under the following GitHub repository: https://github.com/fpiumi/A549_cDNA_Y2H_library_characterization_workflow.

## Results

### Assessment of the sequencing quality of the A549 cDNA library

The vector backbones pDEST22 containing the cDNA inserts were linearized and directly sequenced using long-read technology (Oxford Nanopore Technologies). After basecalling, and the merging of the two sequencing datasets, a total of 1,237,849 reads were generated with 99.5% of the reads above the phred quality score of 10 (Q10) and 52.2% of the reads above Q15 (Fig 2A). The median read quality score is 15.1, indicating a probability of incorrect base call of 1–50 and a base call accuracy of 95% (Table 1). Regarding the read length and read length N50, we see a distribution from 1,000 base pairs (bp) to 10,000 bp (Fig 2B), with a N50 of 7,943 bp and a mean of 6,960.7 bp (Table 1).

Overall, we observe a sequenced read length shorter but close to the expected pDEST22 vector length of 8,930 bp with a satisfying read quality for gene identification, the main purpose of our study.

### Selection of reference sequence dataset for read assignment

We investigated which alignment parameters would be the best to identify these cDNA inserts, corresponding to the possible protein tested in downstream PPI assays. The choice of the reference, human genome or transcriptome with different parameters generates a different number of primary, secondary and supplementary alignments. The primary alignment is the best alignment for a read, as determined by the aligner based on a scoring algorithm (e.g., highest alignment score, least mismatches). A secondary alignment represents an alternative alignment for a read, where the read maps to another location in the reference sequence but is not considered the best alignment. A supplementary alignment is used when a read spans large structural variations (e.g., translocations, inversions) or when it maps in a split manner across multiple regions of the reference. These alignments represent fragments of a single read that align to different parts of the reference. As the goal was to identify cDNAs, four different alignments were tested: genome, genome with splice parameter, transcriptome and transcriptome combined with non-coding RNA annotation and compare the different outcomes (see Materials and Methods section). Of the 1,237,849 reads, the genome mapping results in the most primary

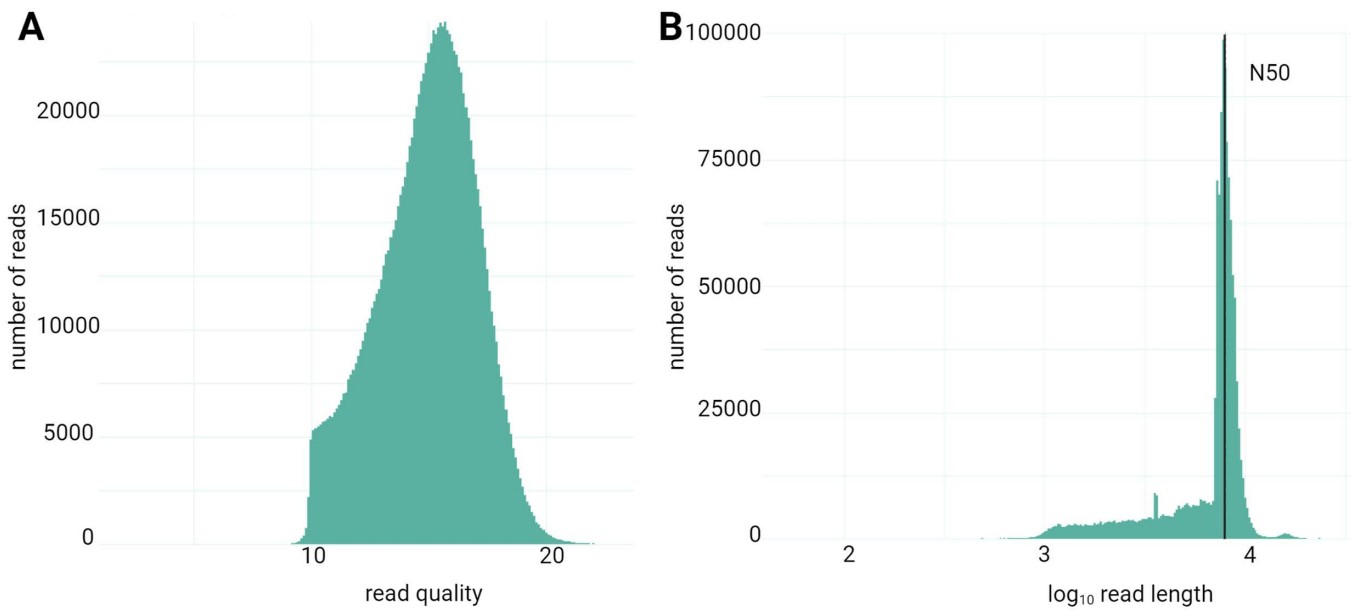

**Fig 2. General statistical analysis of sequencing outputs. A) read quality (nanoplot), B) read length and N50 (nanoplot).**

**Table 1. NanoPlot results.**

| Category | Output |
|---|---|
| Mean read length (bp) | 6,960.7 |
| Min read length (bp) | 61.0 |
| Max read length (bp) | 372,235.0 |
| Mean read quality | 13.9 |
| Median read length (bp) | 7,684.0 |
| Median read quality | 15.1 |
| Number of reads | 1,237,849.0 |
| Read length N50 (bp) | 7,943.0 |
| STDEV read length (bp) | 3,037.2 |
| Total bases | 8,616,303,424.0 |

Statistical output of ONT sequencing of the A549 cDNA library generated with NanoPlot on all 1,237,849 reads generated with the two rounds of sequencing.

alignments (960,335 representing 77.58% of the total reads) and the transcriptome the least, with 868,972 primary alignments, only 70.20% (Fig 3A, Table 2). When looking at the distribution of primary, secondary and supplementary alignments, the genome splice alignment gets the greatest proportion of primary mapped alignments. On the contrary, the genome alignment receives the largest share of supplementary alignments (Fig 3B). Interestingly, when investigating the details of the supplementary alignments with the genome, they mostly map to the same chromosome, same strand and non-overlapping. With the genome splice, we get a similar result than with the transcriptome mapping, most supplementary alignments map to a different chromosome. Importantly, the genome mapping resulted in 1,123,699 supplementary alignments and the genome splice in 162,319 ones. The genome splice mapping allows a better identification of the reads and reduces the supplementary alignments tenfold. A mapping that minimizes the supplementary alignments appears to

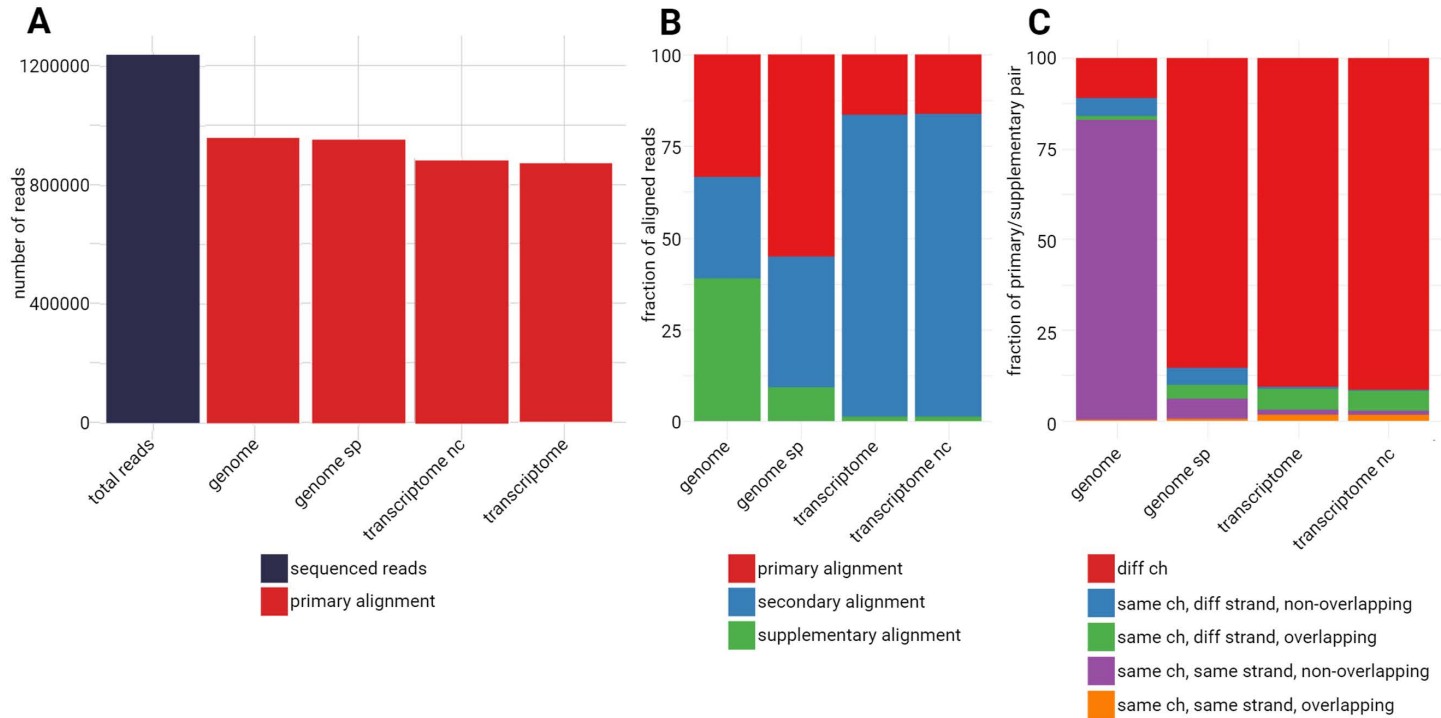

**Fig 3. Characterization of aligned reads. a) Total number of sequenced reads and number of reads with a primary alignment using minimap2 and different human reference genome or transcriptome, with different parameter (see materials and methods section) b) distribution of alignments (primary, secondary, supplementary) according to the used reference c) Investigation of supplementary alignments.** Supplementary alignments are categorized whether they are on the same chromosome, same strand and overlap the primary alignment or not. Ch: chromosome; diff: different; genome sp: genome splice; transcriptome nc: transcriptome + non-coding RNA annotations.

**Table 2. Flagstat results of the different alignments of the ONT generated long reads to different references.**

|  | Genome | Genome splice | Transcriptome | Transcriptome + Non coding |
|---|---|---|---|---|
| minimap2 parameters | *-ax map-ont -N 100* | *-ax splice –N 10* | *-ax map-ont -N 100* | *-ax map-ont -N 100* |
| total | 3,152,897 | 2,017,793 | 5,628,226 | 5,796,734 |
| primary | 1,237,849 | 1,237,849 | 1,237,849 | 1,237,849 |
| secondary | 791,349 | 617,625 | 4,318,361 | 4,478,617 |
| supplementary | 1,123,699 | 162,319 | 72,016 | 80,268 |
| mapped | 2,875,383 (91.20%) | 1,732,658 (85.87%) | 5,259,349 (93.45%) | 5,445,337 (93.94%) |
| primary mapped | 960,335 (77.58%) | 952,714 (76.97%) | 868,972 (70.20%) | 886,452 (71.61%) |
| unique gene names | 17,380 | 16,776 | 12,748 | 13,719 |
| unique transcript IDs | NA | NA | 42,283 | 45.580 |
| Sum final counts | 2,169,516 | 1,164,190 | 456,525 | 476,012 |
| Protein coding | 12,354 | 12,132 | 12,378 | 12,133 |

All aligned reads were marked QC pass.

remove ambiguities in the gene identification of this study. Indeed supplementary alignments arise from reads that cannot be mapped continuously on the reference or from chimeric reads [29]. For these reasons, further analysis was carried out with the genome splice alignment results.

The total number of alignments in the input BAM of the genome mapping with splice parameters is 2,017,793, all of them are marked by samtools flagstat as passing quality controls (QC pass) (Table 2). Of those, 1,732,658 were mapped, 85.87% of the total 2,017,793 alignments, and among those mapped alignments, 952,714 were primary mapped, which represents 76.97%.

### Gene identification

To identify the cDNA inserts composing the A549 cDNA library, the generated reads with ONT sequencing were mapped to the human genome with a splice argument using the minimap2 algorithm (Table S1). The gene distribution is presented in Fig 4A, with 3,323 different genes identified with at least one read in the sequencing data. Most genes have a low count-number, 50% of genes have a count number of less than 11 reads with at the end of the distribution one gene having itself a read count of 22,577, out of a total population of 16,770 genes.

When looking at the distribution of gene types of the identified genes after mapping of the sequenced reads to the spliced genome, a majority of protein coding arose, with 12,123 different genes. Then a total of 4,647 genes, were identified as other gene types such as processed pseudogenes, transcribed unprocessed pseudogenes, long non-coding RNAs (lncRNAs), unprocessed pseudogenes, transcribed processed pseudogenes, rRNA pseudogenes, transcribed unitary pseudogenes (Fig 4B). We only represent gene types with a count over 2 in Fig 4B (Table S2). It is of particular interest that the majority of the A549 cDNA library encodes for protein coding gene, as this cDNA library is used for Y2H and protein-protein interaction studies.

To evaluate the completion of the sequencing of the A549 cDNA library for its comprehensive description, we performed a rarefaction or saturation curve, plotting the number of uniquely identified genes to the number of sequenced reads.

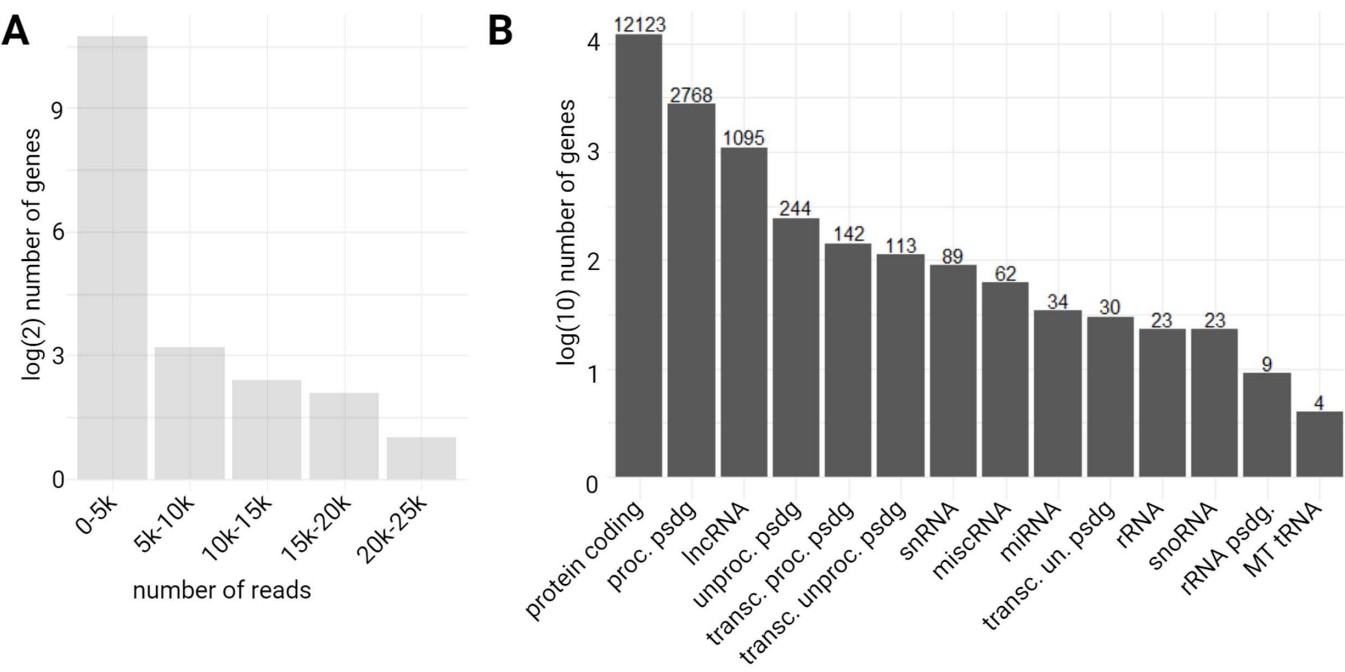

**Fig 4. Mapping results. A) Distribution of the number of reads per number of different genes identified B)** Gene types distribution of mapped reads (proc. psdg: processed pseudogene; lncRNA: long non coding RNA; unproc. psdg: unprocessed pseudogene; transc. proc. psdg: transcribed processed pseudogene; transc. unproc. psdg: transcribed unprocessed pseudogene; snRNA: small nuclear RNA; miscRNA: miscellaneous RNA; miRNA: microRNA; transc. un. psdg: transcribed unitary pseudogene; rRNA: ribosomal RNA; snoRNA: small nucleolar RNA; rRNA psdg: ribosomal RNA pseudogene; MT tRNA: mitochondrial transfer RNA).

An exponential curve was obtained with an approaching plateau. This shows a rapid increase in gene discovery with the first 500,000 sequenced reads and a diminishing discovery rate beyond 1,000,000 sequenced reads. Indeed almost 15,000 different genes were identified with 500,000 reads. Doubling the number of sequenced reads to 1,000,000 does not double the number of unique genes but only yields approximately 16,000 total unique genes (Fig 5). This can also be seen with the nodes calculated to plot this saturation curve (Table S3), first 2,477 genes are found, then it drops to 1,394 and 939, and from the 7th to 10th iterations, less than 500 new genes are added each time with the increase in total sequenced reads (Table S3). From the curve projection (dotted line, Fig 5), we can infer that doubling of total reads would not significantly increase the number of unique genes discovered.

Finally, in order to compare the quality and complexity of the A549 cDNA library from this study, we compared the sequencing results with transcriptomic analysis of A459 cells, using cDNA and ONT sequencing reported by Chen *et al* [30]. The fastq files reported in this study were downloaded and processed as described in the material and methods section, using minimap2 and mapping on the genome using the splice parameter (Table S4). The mapping results demonstrate an average of 11,240 protein coding genes identified in transcriptomics experiments of A549 cells, placing the A549 cDNA library that we describe above average with 12,123 protein coding genes (Table 3). It appears that even with all the cloning process, from the RNA extraction of the A549 cells to the prey plasmids ready for Y2H studies, the A549 cDNA library is representative of the RNA content of A549 cells.

## Functional analysis of the A549 cDNA content

The sequenced A549 cDNA library contains 12,123 protein coding genes. This data set should be representative of a human cell mRNA content. To characterize proteins coded in this cDNA library, we performed functional analysis using bioinformatics tools, exploring databases such as Gene Ontology (GO) or Reactome. Firstly, with a GroupGO analysis, we generated the GO level 2 biological process (BP), cellular compartments (CC) and molecular function (MF) terms, associated with our gene list (Fig 6, Table S5).

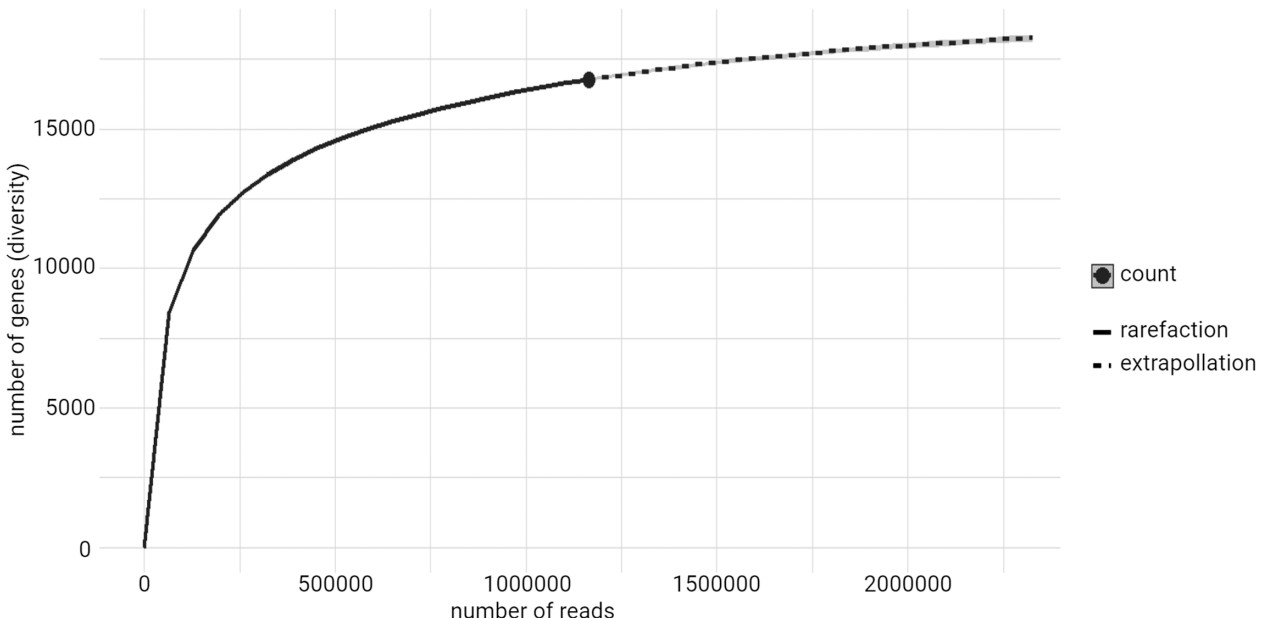

**Fig 5. A549 cDNA library complexity saturation curve depicting the number of unique genes detected with the increase of reads sequenced.** Figure generated with the iNEXT package using the total count table of mapped genes (Table S1).

**Table 3. Comparison of flagstat results of the different Nanopore datasets of cDNA sequencing of A549 cells processed using the same mapping.**

|  | cDNA replicate1 run2 [30] | cDNAStranded replicate3 run3 [30] | cDNAStranded replicate5 run2 [30] | directcDNA replicate3 run1 [30] | This study |
|---|---|---|---|---|---|
| total | 4,849,199 | 3,641,981 | 16,369,074 | 154,818 | 2,017,793 |
| primary | 2,976,193 | 2,429,224 | 12,232,845 | 78,528 | 1,237,849 |
| secondary | 1,824,724 | 1,158,111 | 4,043,898 | 51,267 | 617,625 |
| supplementary | 48,282 | 54,646 | 92,331 | 25,023 | 162,319 |
| mapped | 4,485,282 (92.50%) | 2,940,744 (80.75%) | 9,844,991 (60.14%) | 144,985 (93.65%) | 1,732,658 (85.87%) |
| primary mapped | 2,612,276 (87.77%) | 1,727,987 (71.13%) | 5,708,762 (46.67%) | 68,695 (87.48%) | 952,714 (76.97%) |
| unique gene names | 16,826 | 15,927 | 18,933 | 9,308 | 16,770 |
| Sum final counts | 3,283,009 | 2,044,283 | 6,805,134 | 107,413 | 1,164,190 |
| Protein coding | 12,245 | 11,865 | 13,291 | 7,560 | 12,123 |

Datasets used from [30].

We get twenty-two different BP terms with our gene set. Over 10,000 genes are associated with "cellular process" term, then between 5,000 and 8,000 with "metabolic process", "biological regulation", "regulation of biological process" and "response to stimulus". These are generic BP GO terms and comprise most of the genes from our whole gene set comprising 12,123 protein coding genes. Of interest in host-pathogen studies, the term "immune system process" is also represented with 1,517 genes in our data set, a term composed of 3,057 genes in total.

Two CC terms arise from the group GO analysis: "cellular anatomical entity" and "protein-containing complex". Almost all our 12,123 protein-coding genes are categorized under the label "cellular anatomical entity". The latest term regroups proteins that were identified as interacting in a stable assembly of at least two macromolecules.

Finally, thirty MF terms from our gene set group GO analysis represent many basic cellular functions, with in order of gene count associated, "binding", "catalytic activity", "transcription regulator activity", "molecular adaptor activity", "transporter activity", "ATP-dependent activity" and "translation regulator activity" for example. All these molecular functions could benefit a virus needing to replicate itself in the target host.

Next, to gain insight into the functional relevance of the identified genes from the ONT sequencing data, we also performed enrichment analysis with the human genome as a background reference (Fig 7). There are 1,695 enriched biological process (BP) GO terms (Table S6) representing several cell metabolism activities, such as "ribonucleoprotein complex biogenesis", "RNA splicing", "mitochondrion organization", "establishment protein localization to organelle" as the enriched terms with the lowest p.adjust among the first 20 terms (Fig 7). Regarding cellular components (CC) GO terms, a total of 390 terms are enriched (Table S6) and the first twenty with the lowest p.adjust values are represented in Fig 7. These CC GO terms represent a variety of cellular components such as the nucleus with terms like "chromosomal region", "nuclear speck", "nuclear envelope", the mitochondria 288 with "mitochondrial protein-containing complex", as well as the cytoplasm compartment with "lysosomal membrane". Among molecular function (MF) GO terms, we also observe a variety of terms totaling 358 terms (Table S6), with terms related to RNA processing with the term "catalytic activity, acting on RNA", to DNA processing with "catalytic activity, acting on DNA", protein specific metabolism "ubiquitin-like protein binding" and signal transduction "protein serine kinase activity" or "GTPase regulator activity". The genes identified in the A549 cDNA library are part of many different GO terms representing key cellular activities.

With over 15,000 genes identified, numerous GO terms are highlighted. We chose a simplified tree map representation of the BP GO terms, that reduces redundant terms, enabling the use of broader labels and keywords to categorize functions identified in the sequenced data set (Fig 8, Table S7). Analysis of the simplified tree map reveals five major enriched categories of biological processes. These categories are: "biogenesis checkpoint phase transition", "establishment Golgi

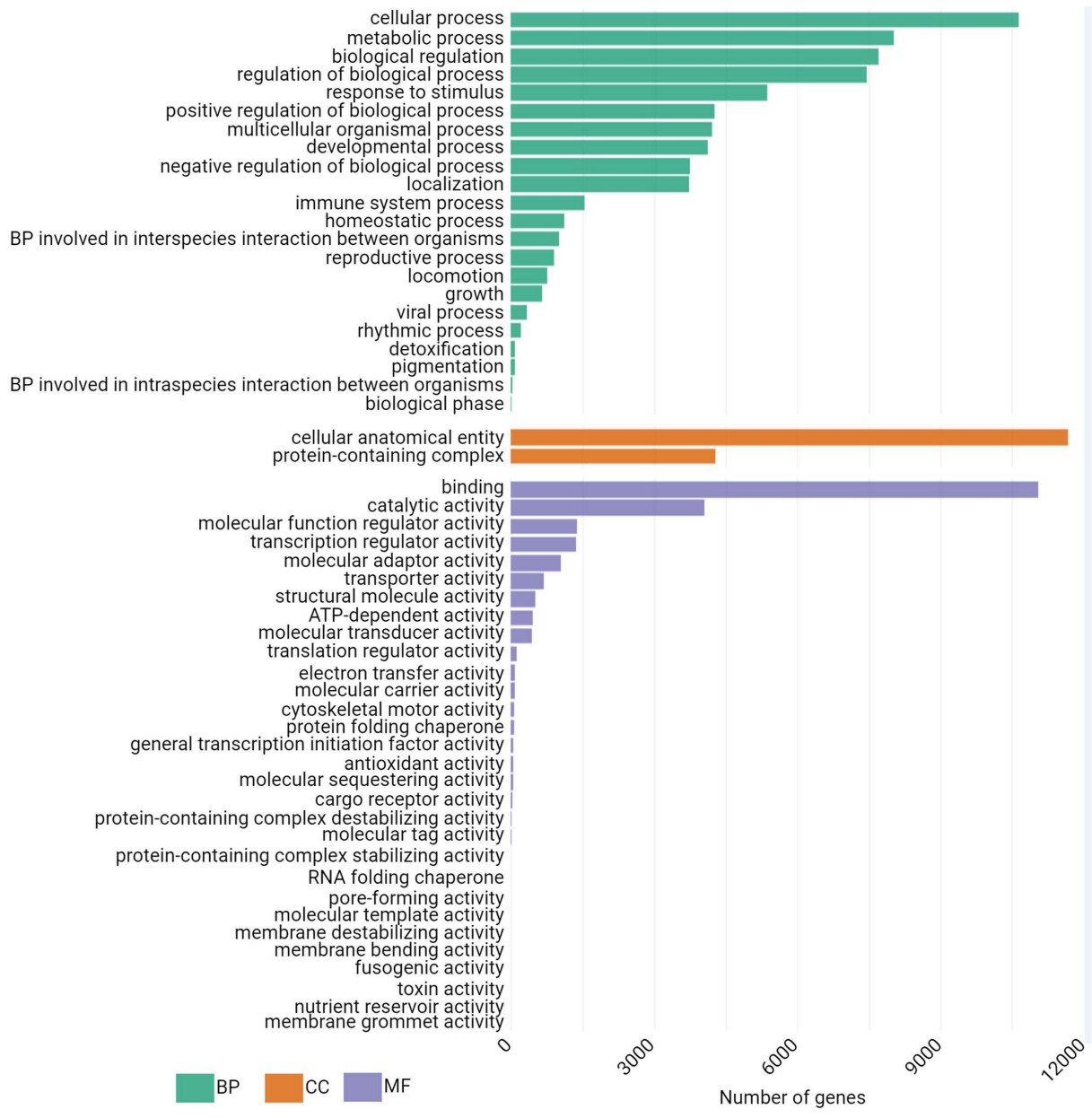

**Fig 6. Group GO analysis with biological process, cellular component and molecular function terms.** The first 20 terms of analysis results are shown; BP = biological process; CC: cellular component; MF: molecular function.

localization transport", "cytoplasmic autophagy assembly catabolic", "double-strand break repair damage "and "rRNA metabolic processing splicing". These five clusters cover multiple localization in the cell such as the cytoplasm, nucleus and Golgi apparatus. Notably, these clusters regroup BP terms that can be implicated in viral infections, possibly hijacked to support viral lifestyle with basic cellular function such as protein translation with Golgi apparatus, processes linked to DNA and RNA as well as autophagy pathways commonly modulated during infections.

Finally, we performed an enrichment analysis using the Reactome pathways database (Fig 9). We obtained 740 pathways (Table S8), displaying here the first 20 with the lowest p.adjust value. Interestingly, Reactome pathways can

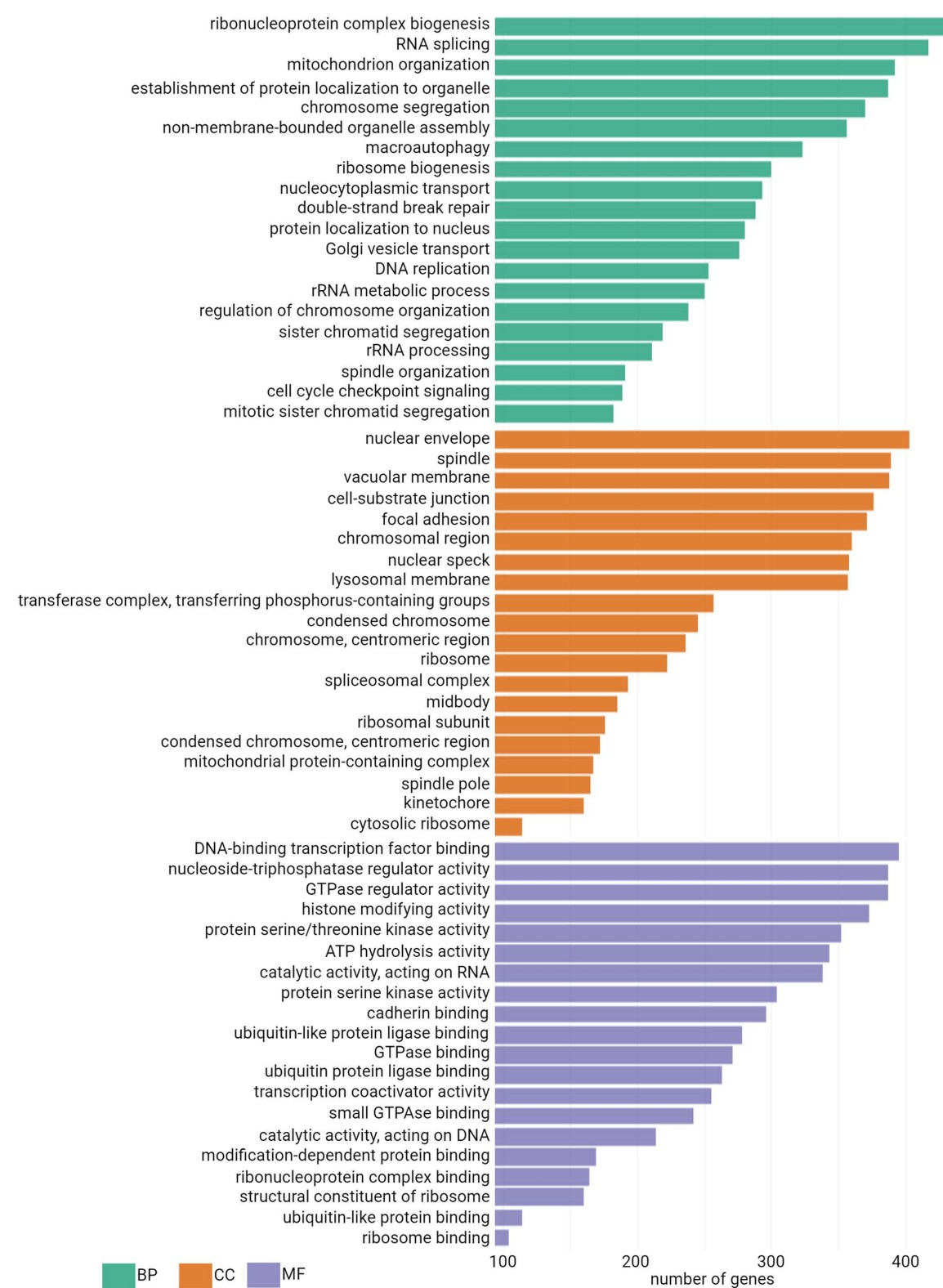

**Fig 7. Gene Ontology (GO) enrichment analysis of biological process, cellular component and molecular function terms associated to the genes sequenced in the A549 cDNA library with the human genome as a background. Thefirst 20 terms for each each, in decreasing order of associated genes are displayed.**

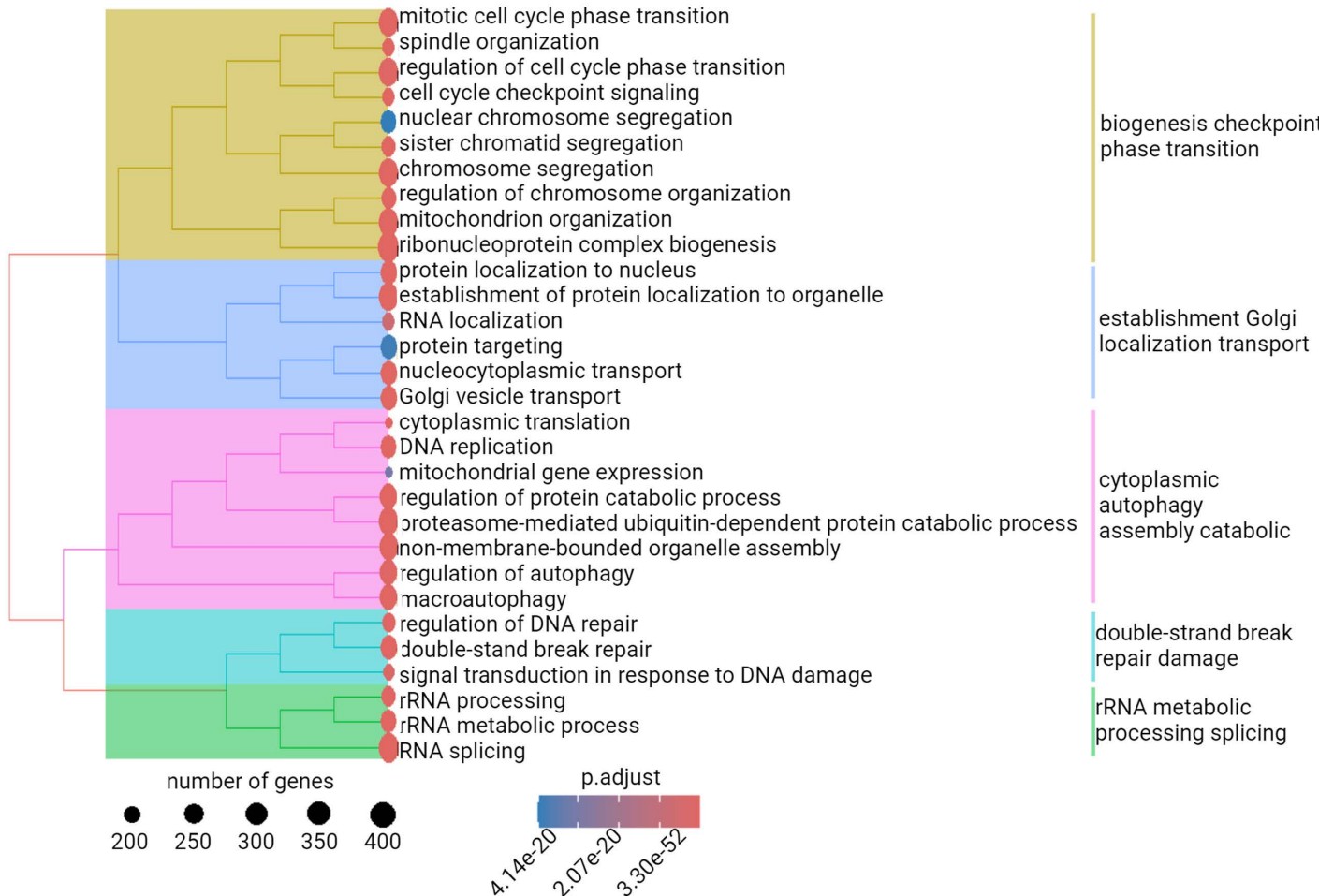

**Fig 8. Simplified tree map representation of enriched Biological Process (BP) Gene Ontology (GO) terms analysis.** Each dot represents a GO term, with size proportional to the number of associated genes and color indicating statistical significance (adjusted p-value < 0.05). Functionally related GO terms are clustered together with an associated color block, based on semantic similarity or hierarchical GO structure.

highlight disease pathways and within the first 20, we find "influenza infection", with 154 genes associated to this pathway in the sequenced dataset. This is of particular interest as we are looking for cellular partners of viral infections with our Y2H screening method. The A549 cDNA library bears genes coding for proteins of many different cellular processes, at different localization of the cell, different timing of the cell cycle and also proteins already identified as important for viral infections. Regarding immunity, we find only one BP, "regulation of type I interferon-mediated signaling pathway", representing 36 proteins in our sequenced dataset (Table S6). Interestingly, the Reactome pathways related to infections are as mentioned above about "influenza", "HIV", but also "SARS-CoV-1" and "SARS-CoV-2" infections, as well as "Parasite infection" and "Bacterial infections". This may reveal to some extent a diversity of proteins associated with a variety of infectious diseases that are of interest for host-pathogen interactions focused studies.

Lastly, as a validation, it is possible to compare the sequencing results of the A549 cDNA library performed in this study, with a previous study of Y2H performed using the same A549 cDNA library to uncover *Trichinella spiralis* NBL1 protein host interactors. All the 20 genes identified with the Y2H assay using this specific A549 cDNA library [31] were indeed sequenced in this present study (Table S9). Both data sets can be represented as a scatter plot (Fig 10), with colonies obtained after the Y2H procedure and the corresponding counts of the sequenced specified gene. When

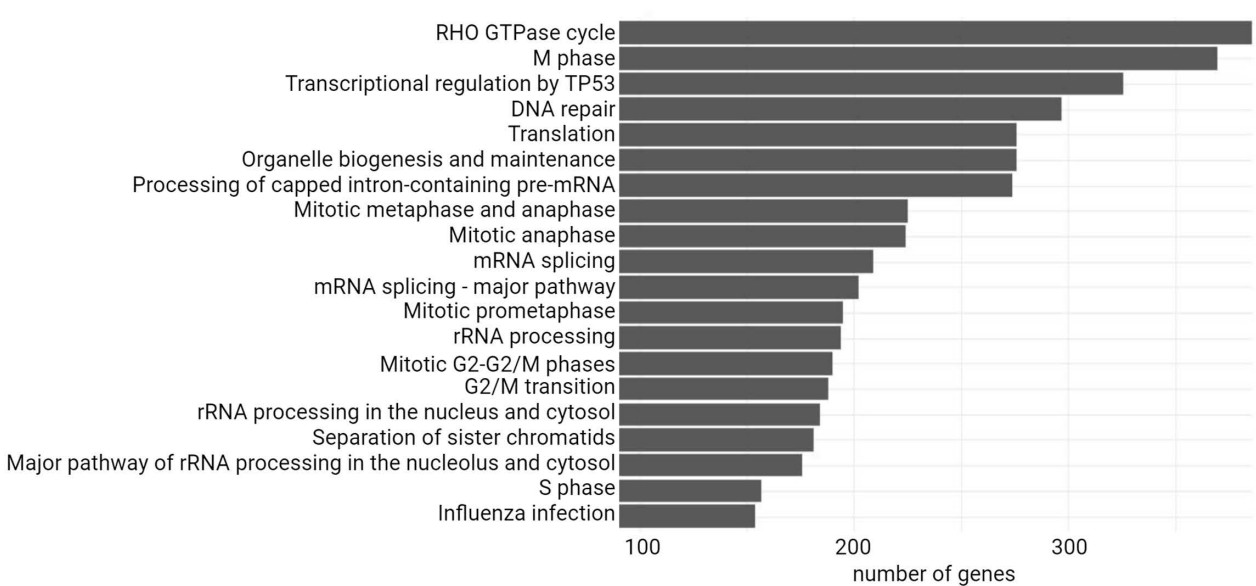

**Fig 9. Reactome pathways enrichment analysis associated to the genes sequenced in the A549 cDNA library with the human genome as a background.** First 20 pathways are displayed, sorted by number of genes associated to the pathway.

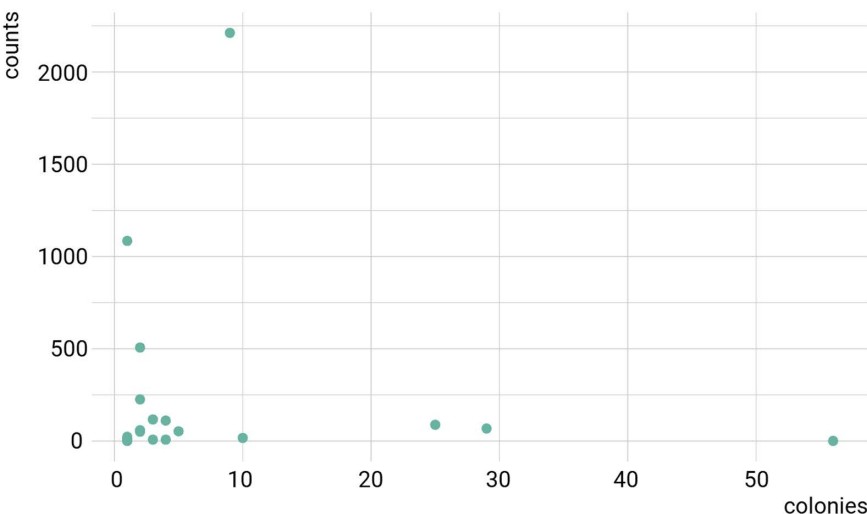

**Fig 10. Scatter plot representation of the number of colonies counted after the yeast-two hybrid procedure of the A549 cDNA library tested against a parasitic protein from *Trichinella spiralis*, and the read counts of the resulting Y2H interactors sequenced in this present study.**

performing a Spearman's rank correlation test, we get a rho = 0.13 with a p-value of 0.57, which indicates there is no correlation between the number of colonies identified with a Y2H assay and the number of counts of the corresponding cDNA sequenced in this study. This precise example seems to point towards the fact that there is no influence on the number of times an interaction is identified with the Y2H assay, and the number of times we identified the same cDNA in the sequencing data (Table S9). This sequencing analysis of the A549 cDNA library adds to the sturdiness of the Y2H assay, as it appears to highlight cellular interactors of the viral proteins, not based on a library bias of over-representation of caught cellular partner, but rather by the relevance of the interaction.

## Discussion

Usual characterization of cDNA library is done through a limited number of clone sequencing or NGS sequencing of PCR-amplified inserts [8]. Here, we report a direct ONT long-read sequencing of linearized Y2H ready vectors with a straightforward bioinformatics pipeline and functional characterization of cDNA library representation.

From the A549 cDNA library, 1,237,849 reads were sequenced, with a median N50 of 7,943 bp and mean length of 6,960.7 bp. The pDEST22 is about 8,900 bp, before recombination with the insert. This is a Gateway cloning system compatible vector, and its resistance cassette, removed with the BP clonase enzyme during the insert cloning step, is about 2,000 bp long. The average length of cloned inserts ranges from 1,200 bp to 1,500 bp, assessed by the subcontractor. It appears that the full-length vectors, backbone with insert, were sequenced as the obtained mean length correlates with the expected value. This helps with downstream analysis, with the estimate of one read equals one insert sequenced. The quality above Q15 for half the reads seems enough for the application of cDNA identification.

By identifying 12,123 protein coding genes in this A549 cDNA library sequencing experiment, we have at least 60% of the expected 20,000 proteins in the human genome [32]. It is important to note that the resulting Y2H ready library we sequenced went through many steps, including insert size selection and cloning. The number of genes expressed of at least one copy per cell in two different cell lines was reported to be between 10,000 and 15,000 [33]. Several available A549 cells ONT sequencing data [30] were bioinformatically processed the same way the data from this study and analyzed for comparison. This revealed that the numbers of unique genes and protein-coding genes found in this study are in the average of other published data.

This analysis does not focus on non-coding genes as this cDNA library is designed to perform Y2H assay. All non-protein-coding genes will not result in the identification of a host factor interacting with a viral protein. We identified a total of 4,647 pseudogenes in our sequencing data. One of the biases induced by cDNA ONT sequencing, as described previously [34] is the identification of pseudogenes rather than the protein coding gene. Here we can consider that identified pseudogenes from our sequencing dataset are most likely protein coding genes, as pseudogenes are not widely expressed. These identified pseudogenes can come from secondary alignments of the primary alignment of the read.

The sequencing saturation curve calculated in this study demonstrates that total completion of the sequencing was not reached. Nevertheless, the doubling of sequenced vectors would not have doubled the discovered unique genes. Importantly, with this sequencing depth, below average of other published data [30], some low abundance transcripts may be missed.

ONT long read sequencing also allowed the use of only one restriction enzyme to linearize the plasmid, being time and cost effective as well as avoiding cutting the inserts of interest. The advantage of this approach lies in minimizing biases potentially introduced during possible PCR steps, ensuring that each sequenced read corresponds directly to a single human gene cloned into the pDEST22 vector. This facilitates the downstream analysis, compared to short reads that would require assembly. The fast-evolving ONT technology, from chemistry to data analysis offers a reliable platform for the cDNA library sequencing application. Indeed, newest R10.4 flowcells show a 10% average increased accuracy compared to R9.4.1 even if GC homopolymers are still difficult to resolve due to challenges linked to the ionic current measurements of such regions [35]. Troubles in resolving repetitive regions can be overcome thanks to analysis tools, even though inherent to the technique long reads are more adapted to the resolution of repetitive regions compared to short reads. They are more easily mapped to a reference compared to the same regions covered by short reads creating ambiguities [16,36]. Regarding the higher error-rate of ONT sequencing, compared to short-read sequencing, this leads to a lack of precision regarding read alignment in particular for splice junctions. Using the right bioinformatics tools developed for long-read processing allows to deal with this issue and does not interfere with transcript quantification [30].

The functional enrichment analysis indicates that we have a representative pool of cellular functions, encompassing basic cellular metabolism, structural roles, and processes specific to viral infections. The latest are of particular interest in the context of Y2H screening for host-pathogen study.

This is an exploratory study of the content of input libraries, an often neglected "black-box" of protein-protein interaction studies. This sequencing solution can help troubleshooting with a high rate of false negatives, inherent to the Y2H method. Indeed, with the knowledge of input screened genes, the non-interaction of an expected interactor could be linked to its absence, wrong isoform or impossibility to screen due to the Y2H assay constraints [37,38].

With this picture of the content of the cDNA library, it would be easy to enrich this library with genes of interest, such as genes involved in the interferon pathway, massively involved in viral infections. If these genes of interest were not found in the sequencing data, a good strategy would be to add the individual vector to this library cDNA pool.

In conclusion, we strongly support the upstream sequencing analysis of assay material, here the prey cDNA library for Y2H screening before further testing. Although some reports suggested NGS use for PPIs discovery using the Y2H assay [14], we believe it is more cost-effective to use sequencing technology once, to characterize the cDNA library, then perform Y2H assay as reported [39].

## Supporting information

**Fig S1. Experimental and analytical steps leading to the characterization of the A549 cDNA library by long-read sequencing.**
(TIF)

**Table S1. Human genome splice mapping of sequencing results count table.**
(XLSX)

**Table S2. Total gene types results.**
(XLSX)

**Table S3. iNEXT results table.**
(XLSX)

**Table S4. Human genome splice mapping of Chen et al 2025 nanopore cDNA datasets.**
(XLSX)

**Table S5. GroupGO complete results.**
(TXT)

**Table S6. EnrichGO complete results.**
(TXT)

**Table S7. Simplified tree map results.**
(TXT)

**Table S8. Reactome results.**
(TXT)

**Table S9. Wang et al 2022 identified proteins comparison table to this study results.**
(XLSX)

## Acknowledgments

We are grateful to the genotoul bioinformatics platform Toulouse Occitanie (Bioinfo Genotoul [40]) for providing computing resources. We thank Dr. Pierre-Olivier Vidalain (CIRI, Lyon, France) for providing the A549 cDNA library. Figures were created with Biorender.com.

## Author contributions

**Conceptualization:** Cécile SCHIMMICH, Mathilde GONDARD, Damien VITOUR, Francois PIUMI.

**Formal analysis:** Cécile SCHIMMICH, Francois PIUMI.

**Funding acquisition:** José-Carlos VALLE-CASUSO, Damien VITOUR.

**Investigation:** Cécile SCHIMMICH, Mathilde GONDARD, Francois PIUMI.

**Methodology:** Cécile SCHIMMICH, Mathilde GONDARD, Francois PIUMI.

**Project administration:** José-Carlos VALLE-CASUSO, Damien VITOUR, Francois PIUMI.

**Supervision:** José-Carlos VALLE-CASUSO, Damien VITOUR, Francois PIUMI.

**Visualization:** Cécile SCHIMMICH, Francois PIUMI.

**Writing – original draft:** Cécile SCHIMMICH, Francois PIUMI.

**Writing – review & editing:** Cécile SCHIMMICH, Mathilde GONDARD, Grégory CAIGNARD, José-Carlos VALLE-CASUSO, Damien VITOUR, Francois PIUMI.

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
