## [Decision Letter · Decision Letter 0]

10 Mar 2025

PONE-D-25-01643Host-Pathogen Protein Interaction Studies: Quality Control of cDNA Libraries Using Nanopore SequencingPLOS ONE

Dear Dr. PIUMI,

Thank you for submitting your manuscript to PLOS ONE. After careful consideration, we feel that it has merit but does not fully meet PLOS ONE’s publication criteria as it currently stands. Therefore, we invite you to submit a revised version of the manuscript that addresses the points raised during the review process.

The manuscript has been evaluated by two reviewers, and their comments are available below.

The reviewers have raised a number of concerns that need attention. They suggest adding more detail to the introduction and better highlighting the strengths and weaknesses of your technique in your Discussion. They also suggest creating a bioinformatics workflow and example dataset to share your code, and request additional information on methodological aspects of the study.

Could you please revise the manuscript to carefully address the concerns raised?

We look forward to receiving your revised manuscript.

Kind regards,

Sarah Jose, Ph.D.

Staff Editor

PLOS ONE

Journal Requirements:

4. Please include captions for your Supporting Information files at the end of your manuscript, and update any in-text citations to match accordingly. Please see our Supporting Information guidelines for more information: http://journals.plos.org/plosone/s/supporting-information .

Reviewers' comments:

Reviewer's Responses to Questions

**Comments to the Author**

1. Is the manuscript technically sound, and do the data support the conclusions?

Reviewer #1: Partly

Reviewer #2: Yes

2. Has the statistical analysis been performed appropriately and rigorously? 

Reviewer #1: N/A

Reviewer #2: Yes

3. Have the authors made all data underlying the findings in their manuscript fully available?

Reviewer #1: Yes

Reviewer #2: Yes

4. Is the manuscript presented in an intelligible fashion and written in standard English?

Reviewer #1: Yes

Reviewer #2: Yes

5. Review Comments to the Author

Reviewer #1: The manuscript describes a method to test completeness or cDNA Y2H libraries using Nanopore sequencing. The authors applied the method to a cDNA library from A549 human lung carcinoma cells, tested the accuracy across multiple references and identified 12,123 protein-coding genes. This approach ensures the completeness of cDNA libraries before Y2H screening, thereby improving the accuracy and reliability of PPI studies.

Despite the authors claim the importance of the cDNA characterization, there are several aspects that were not tested nor presented to the community, which I would recommend including to support their study. First, one aspect that was not mentioned in the study consists of the identification of the translation frame of the identified fragments. When prey fragments are fused tot eh prey vector, different translation frames can be generated. I recommend to the authors to clarify if the preys are complete CDS sequences or if they are fragments, and if all are expected to be in frame with the prey or if partial prey fragments are present and what frame they are in. This is a very important aspect to identify false negatives and false positives in the Y2H assay, as out of frame fragments could interfere with the interaction identification.

Second, if authors are presenting a method that will ensure reliability of Y2H studies, they should present the code and even a bioinformatic workflow to the scientific community. I recommend to present the analysis workflow in github with an example dataset of the analyses that were performed in the study.

Lastly, in the discussion authors mentioned a validation dataset observing no correlation between the number of times an interaction is identified with the Y2H assay, and the number of times we identified the same cDNA in the sequencing data. I think this is a very important observation that needs to be expanded. A more through analysis, properly described in the results section is required to validate method and identify useful metrics that can inform Y2H assays. For example, is there a minimum number of nanopore reads that represent preys that can be identified in Y2H, considering aspects as genetics drift, bottlenecks and positive selection in the yeast population during the Y2H assay.

Reviewer #2: This study innovatively applies Oxford Nanopore Technologies long-read sequencing to quality control of yeast two-hybrid (Y2H) cDNA libraries, addressing the limitations of conventional methods in assessing library coverage and functional representation. The experimental design integrates sequencing saturation curve analysis for library completeness evaluation with pathway enrichment validation to confirm biological relevance, providing novel technical perspectives for host-pathogen interaction research. While the overall experimental framework (covering library construction, sequencing, data analysis, and functional validation) is generally sound, the following issues require improvement:

1.Please add some content regarding the application of Nanopore Sequencing Technology in quality control of cDNA libraries in the introduction section.

2.Some of the figure explanations are not clear enough. Please add some figure explanations more detailed alongside the figures, such as Figures 5 and 7-9.

3.The quality of RNA, efficiency of cDNA synthesis, and size distribution of inserted fragments in library construction are not clearly defined.

4.12123 protein-coding genes were detected, comparative analysis with existing Y2H library studies should be supplemented to contextualize these findings.

5.A direct comparative analysis of gene detection performance between Nanopore Sequencing Technology and Illumina platforms using identical libraries is essential to substantiate the technical advantages of long-read sequencing.

6.The discussion should address inherent limitations of Nanopore Sequencing Technology technology, such as the coverage bias in high-GC or repetitive genomic regions, and the inability to distinguish alternative splicing isoforms.

7.The potential impact of undetected coding genes on Y2H false-negative rates should be systematically discussed.

6. PLOS authors have the option to publish the peer review history of their article (what does this mean? ). If published, this will include your full peer review and any attached files.

**Do you want your identity to be public for this peer review?** For information about this choice, including consent withdrawal, please see our Privacy Policy .

Reviewer #1: No

Reviewer #2: No

---

## [Author Response · Author response to Decision Letter 1]

18 Apr 2025

Dear Sir/Madam,

Many thanks to you for your helpful feedback. We address the reviewers concerns to the best of our abilities. We think that these remarks improved the manuscript and we are grateful for the editor’s considerations as well as the time the reviewers took to formulate their reviews. Below, the responses are following the points raised.

Reviewer #1: The manuscript describes a method to test completeness or cDNA Y2H libraries using Nanopore sequencing. The authors applied the method to a cDNA library from A549 human lung carcinoma cells, tested the accuracy across multiple references and identified 12,123 protein-coding genes. This approach ensures the completeness of cDNA libraries before Y2H screening, thereby improving the accuracy and reliability of PPI studies.

Despite the authors claim the importance of the cDNA characterization, there are several aspects that were not tested nor presented to the community, which I would recommend including to support their study.

Thank you for this comment, we included your suggestions the best we could.

First, one aspect that was not mentioned in the study consists of the identification of the translation frame of the identified fragments. When prey fragments are fused tot eh prey vector, different translation frames can be generated. I recommend to the authors to clarify if the preys are complete CDS sequences or if they are fragments, and if all are expected to be in frame with the prey or if partial prey fragments are present and what frame they are in. This is a very important aspect to identify false negatives and false positives in the Y2H assay, as out of frame fragments could interfere with the interaction identification.

The library construction was subcontracted to Thermo Fisher Scientific, using a custom uncut three frame cloning method. It is specified that the cDNAs were ligated with “custom three reading frame adaptors containing Shine Delgarno and Kozak sequences at the 5’ ends”. Regarding if the cDNAs are full CDS or not, they underwent an insert size selection, >1kb. The construction of cDNA libraries is carried out on such a scale that a control of every cDNA is not possible. Nevertheless, our sequencing study enables us to investigate this question in more detail, but on a case specific basis. With this dataset, it is easy to look up for a specific identified gene, and check if it codes for the full-length version of it. If not, this can help to identify the interacting part of the protein useful in the PPI, before further site studies after PPI identification.

Second, if authors are presenting a method that will ensure reliability of Y2H studies, they should present the code and even a bioinformatic workflow to the scientific community. I recommend to present the analysis workflow in github with an example dataset of the analyses that were performed in the study.

Thank you for the suggestion! Please check out https://github.com/fpiumi/A549_cDNA_Y2H_library_characterization_workflow

We added this information in the materials and methods section as well. The example dataset is the one from the study, already available on a public repository, identified within the materials and methods section: NCBI, accession number: SRR31585952 https://www.ncbi.nlm.nih.gov/sra/SRR31585952

Lastly, in the discussion authors mentioned a validation dataset observing no correlation between the number of times an interaction is identified with the Y2H assay, and the number of times we identified the same cDNA in the sequencing data. I think this is a very important observation that needs to be expanded. A more through analysis, properly described in the results section is required to validate method and identify useful metrics that can inform Y2H assays. For example, is there a minimum number of nanopore reads that represent preys that can be identified in Y2H, considering aspects as genetics drift, bottlenecks and positive selection in the yeast population during the Y2H assay.

Thank you for this comment. We moved the comparison with Wang et al paper to the results section. Unfortunately, as this study only identified 20 proteins and that it is the only published work using this cDNA library, there is no statistically significant relationship between the sequenced counts and the number of times an interaction is uncovered in the yeast. We added a scatter plot and calculated a Spearman’s rank correlation test, both demonstrating that there is no correlation between the two datasets. This comparison can only give a small insight into the fact that there appears to be no correlation between cDNA counts and the number of times an interaction can be identified in a yeast-two hybrid assay. We cannot provide a robust relationship between a minimum number of nanopore reads corresponding to preys that could be identified in downstream Y2H assay. Genetics drift, bottlenecks and positive selection within the yeast population during a Y2H assay are inherent to the technique and the sequencing of the cDNA library partner is a way to maximize knowledge and control over the assay.

Reviewer #2: This study innovatively applies Oxford Nanopore Technologies long-read sequencing to quality control of yeast two-hybrid (Y2H) cDNA libraries, addressing the limitations of conventional methods in assessing library coverage and functional representation. The experimental design integrates sequencing saturation curve analysis for library completeness evaluation with pathway enrichment validation to confirm biological relevance, providing novel technical perspectives for host-pathogen interaction research. While the overall experimental framework (covering library construction, sequencing, data analysis, and functional validation) is generally sound, the following issues require improvement:

Thank you for your comment, we have tried to improve our manuscript according to your recommendations.

1.Please add some content regarding the application of Nanopore Sequencing Technology in quality control of cDNA libraries in the introduction section.

There is no literature available on the use of Nanopore Sequencing Technology for quality control of cDNA libraries used for applications such as yeast two-hybrid or any genomic or proteomic testing, with a full count table and functional analysis. Nevertheless, the already mentioned references around this topic were discussed in more detail. Some references were added regarding this point, to highlight the fact that this study is the first of its kind, underlining the importance and primacy of this study.

2.Some of the figure explanations are not clear enough. Please add some figure explanations more detailed alongside the figures, such as Figures 5 and 7-9.

To gain in clarity, we extended both captions and in-text explanations.

3.The quality of RNA, efficiency of cDNA synthesis, and size distribution of inserted fragments in library construction are not clearly defined.

This cDNA library construction was subcontracted to Thermo Fisher Scientific. We added the maximum details we knew in the “Materials and methods section”.

4.12123 protein-coding genes were detected, comparative analysis with existing Y2H library studies should be supplemented to contextualize these findings.

Such studies are not done of Y2H cDNA libraries, we propose an innovative control of input cDNA libraries in these PPI. Nevertheless, with the recently published benchmark of Nanopore sequencing of human cell lines for transcriptomics by Chen et al, 2025, we were able to compare RNAseq quality cDNA sequencing using an ONT MinION platform with our sequencing experiment of a PPI cDNA library, both from A549 cells. We processed these datasets with the same bioinformatics pipeline described here and added this comparison to the results section. The results of this study frame our data, particularly in terms of the number of unique genes and protein-coding genes identified, and our figures are in the average of these published sequencing data. This finding shows that the Y2H ready A549 cDNA library sequenced in this paper is still representative of A549 cell line RNA content despite all cloning steps. Downstream PPI studies are strengthened, as the starting material is of high quality in terms of potential diversity.

5.A direct comparative analysis of gene detection performance between Nanopore Sequencing Technology and Illumina platforms using identical libraries is essential to substantiate the technical advantages of long-read sequencing.

Thank you for this comment, but the important part of the study we want to highlight is the characterization of a Y2H ready library, and not a benchmark of techniques. We rephrased a bit the way we were presenting this in the discussion, as we want to underline that the advantage of ONT is the generation of long reads, which facilitate bioinformatics downstream processing. It is also a cheaper solution and we are demonstrating that the volume of output data is sufficient, as Illumina is known for its deep sequencing.

6.The discussion should address inherent limitations of Nanopore Sequencing Technology technology, such as the coverage bias in high-GC or repetitive genomic regions, and the inability to distinguish alternative splicing isoforms.

We have taken this comment into account in the discussion section. We extended the discussion around the inherent limitations of the Nanopore sequencing technology.

7.The potential impact of undetected coding genes on Y2H false-negative rates should be systematically discussed.

We already addressed this remark in the discussion. However, as it appears not clear enough, we emphasize this remark in the corrected manuscript, even though this potential impact is an inherent bias of such assays and the sequencing of the cDNA library does not eliminate it. On the contrary, we offer here the ability to remove this potential assay bias with providing a thorough list of known inserts.

Sincerely,

François Piumi, MSc, Research associate, INRAE

---

## [Decision Letter · Decision Letter 1]

4 May 2025

Host-pathogen protein interaction studies: quality control of cDNA libraries using nanopore sequencing

PONE-D-25-01643R1

Dear Dr. Piumi,

We’re pleased to inform you that your manuscript has been judged scientifically suitable for publication and will be formally accepted for publication once it meets all outstanding technical requirements.

Kind regards,

Stephen D. Ginsberg, Ph.D.

Section Editor

PLOS ONE

**Comments to the Author**

1. If the authors have adequately addressed your comments raised in a previous round of review and you feel that this manuscript is now acceptable for publication, you may indicate that here to bypass the “Comments to the Author” section, enter your conflict of interest statement in the “Confidential to Editor” section, and submit your "Accept" recommendation.

Reviewer #1: All comments have been addressed

Reviewer #2: All comments have been addressed

2. Is the manuscript technically sound, and do the data support the conclusions?

Reviewer #1: Yes

Reviewer #2: Yes

3. Has the statistical analysis been performed appropriately and rigorously? 

Reviewer #1: Yes

Reviewer #2: Yes

4. Have the authors made all data underlying the findings in their manuscript fully available?

Reviewer #1: Yes

Reviewer #2: Yes

5. Is the manuscript presented in an intelligible fashion and written in standard English?

Reviewer #1: Yes

Reviewer #2: Yes

6. Review Comments to the Author

Reviewer #1: (No Response)

Reviewer #2: The authors have modified the manuscript according to suggestions. We don't have any additional comments for the author.

7. PLOS authors have the option to publish the peer review history of their article (what does this mean? ). If published, this will include your full peer review and any attached files.

**Do you want your identity to be public for this peer review?** For information about this choice, including consent withdrawal, please see our Privacy Policy .

Reviewer #1: No

Reviewer #2: No

---

## [Editor Report · Acceptance letter]

PONE-D-25-01643R1

PLOS ONE

Dear Dr. PIUMI,

I'm pleased to inform you that your manuscript has been deemed suitable for publication in PLOS ONE. Congratulations! Your manuscript is now being handed over to our production team.

Kind regards,

on behalf of

Dr. Stephen D. Ginsberg

Section Editor

PLOS ONE